# Ultrasonic-Assisted Extraction of Phenolic Compounds from *Lonicera similis* Flowers at Three Harvest Periods: Comparison of Composition, Characterization, and Antioxidant Activity

**DOI:** 10.3390/molecules29143280

**Published:** 2024-07-11

**Authors:** Yunyi Hu, Wenzhang Qian, Shaojun Fan, Yao Yang, Hai Liao, Guoqing Zhuang, Shun Gao

**Affiliations:** 1Department of Forestry, Faculty of Forestry, Sichuan Agricultural University, Chengdu 611130, China; huyunyi@stu.sicau.edu.cn (Y.H.); 202001247@stu.sicau.edu.cn (W.Q.); fanshaojun@stu.sicau.edu.cn (S.F.); yangyao@stu.sicau.edu.cn (Y.Y.); 2School of Life Science and Engineering, Southwest Jiaotong University, Chengdu 611756, China; ddliaohai@home.swjtu.edu.cn; 3Sichuan Academy of Forestry, Chengdu 610081, China

**Keywords:** *Lonicera* *similis*, harvest stages, extraction process, phenolic compounds, antioxidant activity, correlation analysis

## Abstract

*Lonicera similis* Hemsl. (*L. similis*) is a promising industrial crop with flowers rich in phenolic compounds. In this study, an ultrasound-assisted extraction (UAE) was designed to extract phenolic compounds from *L. similis* flowers (LSFs). A contrastive analysis on the phenolic compounds’ yield and characterization and the antioxidant activity of the extracts at three harvest stages (PGS I, PGS II, and PGS III) are reported. The results indicate that the optimal conditions are a sonication intensity of 205.9 W, ethanol concentration of 46.4%, SLR of 1 g: 31.7 mL, and sonication time of 20.1 min. Under these optimized conditions, the TPC values at PGS I, PGS II, and PGS III were 117.22 ± 0.55, 112.73 ± 1.68, and 107.33 ± 1.39 mg GAE/g, respectively, whereas the extract of PGS I had the highest TFC (68.48 ± 2.01 mg RE/g). The HPLC analysis showed that chlorogenic acid, rutin, quercetin, isoquercitrin, and ferulic acid are the main components in the phenolic compounds from LSFs, and their contents are closely corrected with the harvest periods. LSF extracts exhibited a better antioxidant activity, and the activity at PGS I was significantly higher than those at PGS II and PGS III. The correlation analysis showed that kaempferol and ferulic acid, among the eight phenolic compounds, have a significant positive correlation with the antioxidant activity, while the remaining compounds have a negative correlation. Minor differences in extracts at the three harvest stages were found through SEM and FTIR. These findings may provide useful references for the optimal extraction method of phenolic compounds from LSFs at three different harvest periods, which will help to achieve a higher phytochemical yield at the optimal harvest stage (PGS I).

## 1. Introduction

The genus *Lonicera* Linn., within the Caprifoliaceae family, includes about 200 species that are mainly distributed in the temperate and subtropical zones of Asia, North America, Europe, and Northern Africa. China alone has 98 species, and China possesses the majority of species [1]. Like other plants of the genus *Lonicera* L., *Lonicera similis* Hemsl. (*L. similis*), known as Chuan Yin Hua, is a perennial deciduous woody liana with numerous valuable medicinal properties. *L. similis* has garnered significant global attention due to its higher phenolic content, extensive cultivation, prolonged flowering period, and abundant flowers. The utilization of *L. similis*, like *Lonicera japonica* (*L. japonica*), is mainly concentrated on the flower bud and/or flowers, which mainly contained polyphenol, flavonoids, polysaccharide, saponins, and iridoids [2,3,4]. Among these bioactive compounds, chlorogenic acid (CGA) and quercetin are the major bioactive compounds, and their contents are used as the evaluation index of flower quality in *L. japonica* [5,6,7]. These components are closely related to anti-oxidative, antibacterial, anti-diabetic, anti-inflammatory, anticancer, and antiviral properties [8,9]. Thus, the flower of *L. similis* and the other species of *Lonicera* have been widely used in the form of solid drinks, distilled liquid of flowers, and healthy beverages in food and non-food fields due to their various functional components and beneficial health values [10,11]. Moreover, it has been reported that the extracts of the flowers and/or buds can be used to treat severe acute respiratory, arthritis, H1N1 influenza, diabetes, fever and infections, novel coronavirus, and so on [12,13].

It is well known that the synthesis of phenolic compounds in flowers is controlled by many factors, and their developmental stages play an important role in impacting the biosynthesis, concentration, and accumulation of phenolic compounds in various flowers [14]. In *L. japonica* flowers, significant differences in flower color and chemical composition have been recorded at different developmental stages, which affect their industrial quality and their intended application [15,16,17,18]. Flowers collected at the silvery flower stage are primarily applied for essential oil extraction, while those harvested at the slightly white alabastrum stage are predominantly applicable to the pharmaceutical filed [19]. Moreover, developmental variation studies on the content of phenolic compounds of flowers in some plant species, such as *Cercis chinensis*, daylilies, and peach, have also been reported, and significant changes have been recorded [20,21,22]. The phenolic compounds contents and compositions of these flowers varied in a flowering-dependent manner, which is also related to the geographical location and extraction methods [5,19,23]. The organs–developmental stages interaction leads to significant variations in the production of phenolic compounds. These phenomena are probably due to the fact that the biosynthesis of phenolic compounds is usually an adaptive response that is mediated by the chemical interaction between organs and their developmental stages [24,25,26]. Sometimes the harvested flowers may not contain the desired quantity of bioactive compounds to meet the industrial standards or consumers’ demands and are often rejected, causing economic loses. It is not surprising that the optimal harvesting stages of the flowers is very influential in order to obtain a high quality of raw materials and finished products as well as a maximum concentration of bioactive compounds for the availability of more effective drugs, which could affect marketability and the consumer acceptance of the final product and its retail price [27,28]. These studies on the accumulation of bioactive compounds during different developmental stages can provide references for the in-depth utilization and exploitation of flower resources.

Phenolic compounds are a series of natural bioactive constituents in the flowers, leaves, and fruits of plants and are generally divided into phenolic acids and flavonoids. Phenolic acids are found in plant tissues, primarily as hydroxyl derivatives of benzoic and cinnamic acids. Flavonoids are grouped into flavanones, flavanols, flavones, isoflavones, flavonols, and anthocyanins based on their chemical structure. Phenolic compounds not only represent very effective antioxidants in plant tissues, but also are a main raw material widely used in the chemical, pharmaceutical, and food industries [29,30]. Thus, the utilization and further study of these phenolic compounds are essential to design and establish an effective extraction process. A series of extraction techniques, like solid–liquid extraction, heat reflux extraction, microwave-assisted extraction (MAE), ultrasound-assisted extraction (UAE), simultaneous MAE/UAE, and supercritical extraction, have been reported for the extraction phenolic compounds and extracts from various flowers [31,32]. It is difficult to establish a universal process for extracting various phenolic compounds due to the differences in plant matrix, composition of phenolic compounds, and their structural diversities [33]. Among these methods, UAE may produce shock waves and hydrodynamic forces on the material surface and further produce cracks in the plant cell walls, allowing solvents to permeate during the extraction process. More importantly, it can accelerate mass transfer, which will help to release the bioactive compounds from plant materials and more easily flow into the solvent. At present, UAE represents less energy consumption, shorter time, less solvent, less active ingredient damage, higher quality and yield of the product, and a lower cost [34]. For these reasons, UAE has been widely applied to extract phenolic compounds from various flowers, including *L. japonica*, *Osmanthus fragrans*, *Bougainvillea glabra*, and *Limonium sinuatum* [35,36,37,38]. When these specific UAE procedures were developed using response surface methodology (RSM), the extraction parameters, including the ultrasound power, time, solid–liquid ratio (SLR), and solvent concentration, were optimized for each other to extract the maximum quantity of phenolic compounds from the flowers [38,39]. Thus, these procedures of UAE contain various parameters and interactions that can impact the extraction yield and a large range of biological phenolic compounds from plant materials.

Although numerous studies have been reported on the extraction of phenolic compounds from different flowers, phenolic compounds obtained from *L. similis* flowers at different harvest stages due to complex constituents are still worth exploring. To this end, the aim of this study is to elucidate the optimal process for phenolic compounds in *L. similis* flowers using UAE, and the differences in total phenolic and flavonoid compounds, eight phenolic compounds, and the antioxidant capacity at three harvest stages were comparatively analyzed. Moreover, the functional groups and microstructures of the gained extracts were assessed using Fourier transform infrared spectroscopy (FT-IR) and scanning electron microscopy (SEM). This study developed an efficient extraction procedure of phenolic compounds from *L. similis* flowers, which will help to screen the developmental variation in phenolic compounds and inform their optimal harvesting stages in favor of their valorization as an interesting source of natural antioxidants in the food and pharmaceutical industries.

## 2. Results and Discussion

### 2.1. Single-Factor Experiments

As depicted in Figure 1a, the yields of TPC and TFC exhibit an upward trend with increasing power, while the yields remain relatively stable between 200 W and 300 W of ultrasound power. The maximum values of TPC and TFC are 111.48 ± 2.85 mg GAE/g, and 67.69 ± 0.74 mg RE/g when the ultrasonic power is 300 W, respectively. A previous report has demonstrated that an ultrasonic power of 300 W is optimal for extracting flavonoids from *Acanthopanax senticosus* [40]. However, the optimal value of ultrasonic power for extracting phenolic compounds from *Lycium ruthenicum* Murr. fruit was only 100 W. This also might be due to the differences in the functional groups, molecular mass, and polarity of the phenolic compounds from different plant organs [41]. In this study, an ultrasonic power of 300 W was determined as the optimal extraction condition for obtaining phenolic compounds from LSF. Possible mechanisms for sonication include fragmentation, erosion, capillary action, and texture removal. Additionally, ultrasonic waves may improve the contact surface area between the material particle and solvent by disrupting cells and generating microcavities within the solution, thereby facilitating extraction. However, with increasing ultrasonic power, the rise in mechanical vibration and temperature leads to reduced cavitation and lower extraction efficiency [42].

Ethanol and water are widely employed as solvents in industrial extraction processes, and their proportions are related to the extraction yield of bioactive compounds from plants [43]. As exhibited in Figure 1b, the extraction yield of TPC firstly increases and then decreases with the rising ethanol concentration. The yield reached its maximum value of 111.77 ± 4.68 mg GAE/g, and 64.33 ± 1.68 mg RE/g when the ethanol concentration was 50%. This may be due to the fact that a lower ethanol concentration has the potential to damage cell membranes, while a large amount of polar solvents can promote the expansion of plant materials due to their water absorption and the release of solutions [44]. However, with the rising ethanol concentration, the polarity of the solution undergoes changes, which correspondingly impacted the solubility of phenolic compounds and non-phenolic compounds, consequently affecting the extraction efficiency [45,46]. Similarly, a higher content of phenolic compounds is obtained from pomegranate peel at the same concentration [47], and chokeberries under the condition of 50% ethanol concentration [48]. Thus, an ethanol concentration of 50% represents a suitable parameter for the optimal extraction of TPC from LSF.

A reasonable SLR is crucial for the extraction yield of bioactive compounds, which is also vital to minimize energy consumption and operational expenses [49]. As exhibited in Figure 1c, the extraction yields of TPC and TFC first show an increase after the reduction trends with a rising SLR. The maximum yields of TPC (112.26 ± 2.23 mg GAE/g) and TFC (58.15 ± 3.89 mg RE/g) were recorded at an SLR of 1:30. However, a gradual decrease in the yields of TP and TF was observed when further increasing the SLR. Other reports indicated that the TPC yields of saffron and blueberry are significantly related to the SLR [50,51]. This may be due to the dependence of substance solubility on intermolecular or interionic interactions between the solute and solvent. However, with the increase in the SLR, the complete dissolution or release of other substances may impact the dissolution of TPC, which can reduce their extraction efficiency and yield [51]. Thus, the SLR of 1:30 was chosen as the suitable parameter for the optimal extraction of LSF-PC.

As depicted in Figure 1d, it is evident that the yields of TPC and TFC significantly increase with the time extension from 5 min to 20 min. When the extraction time was raised to 30 min, the yield of TFC remarkably reduced (*p* < 0.05). A previous study indicated that the duration of the UAE process is the primary variable influencing the extraction rate of active components. Moreover, a suitable extraction time can improve the diffusion of soluble compounds and the selective extraction of active substances from plants [52]. However, a time that is too long can lead to the reduction in the yields of phenolic compounds and other components [42]. Thus, to raise the extraction efficiency of phenolic compounds, 15 min was selected as the optimal condition for conducting further testing.

### 2.2. Optimization Extraction

The extraction yield is influenced by multiple factors and their interactions, highlighting the need for further exploration and investigation of the impacts of these factors and their interactions on the efficiency of phenolic compounds’ extraction. During this study, the extraction conditions for phenolic compounds were optimized by coding and considering four variables: ultrasonic power, ethanol concentration, SLR, and extraction time. Analysis of variance was used to evaluate the fit of the second-order polynomial equation to the experiment results (Table 1). The quadratic regression model is significant, with a less than 5% critical value, but there is no significant fit (*p* > 0.05), showing that the mathematical model is highly reliable and feasible, and used to calculate TPC and TFC. The coefficients of determination (R^2^) are 0.9472 and 0.9136, indicating that both models have good accuracy. The adjusted coefficients of determination (R^2^_Adj_), 0.8857 and 0.8129, exhibited that the model is highly reliable to predict the tested results. Taking into account the actual feasibility, the optimal UAE parameters were checked as follows: ultrasonic power of 100 W, ethanol concentration of 80%, SLR of 1:30 g/mL, and extraction time of 15 min.

Figure 2a–f exhibit the relationship between extraction parameters and TPC, as well as the interaction between variables using the three-dimensional (3D) surface diagram. In addition, Appendix A lists the experimental and predicted values of the coding levels and TPC yield. In 27 experiments, the highest yield of TPC of 116.90 ± 3.13 mg GAE/g was observed in Group 13, but the lowest TPC of 85.97 ± 3.54 mg GAE/g was observed in Group 19. The effective terms in the model equation obtained by TPC include the linear terms of B, C, and D, and the quadratic terms of A^2^, B^2^, C^2^, D^2^, and BC (Table 1). To better understand how independent variables affected the yield of TPC, the model was established via second-order polynomial quadratic equations, as is shown below.
YTPC = 114.90 − 4.07B − 4.1C + 2.55D − 5.06AD − 6A^2^ − 17.25B^2^ − 6.19C^2^ − 3.66D^2^
(1)
where YTPC represents the yield of TPC. A, B, C, and D represent the four independent variables of ultrasonic power, ethanol concentration, SLR, and extraction time, respectively. According to the coded model (Equation (1)), the difference in the interaction between ethanol concentration (B) and SLR (C) was notable (*p* < 0.05) among four independent variables. As shown in Table 1, the influencing order of extraction conditions on TPC are ethanol concentration > SLR > extraction time > ultrasonic power. It shows that ethanol concentration, SLR, and extraction time are all significant factors affecting the yield of TPC. However, it should be noted that these decreasing effects were evident for both linear and quadratic factors.

Appendix A shows that the ranges of the response values and predicted values of TFC are 47.86–69.22 mg RE/g and 46.66–66.07 mg RE/g, respectively. The F values and *p* values indicated the model’s extreme significance (*p* < 0.01, Table 1). However, the fitting loss of each model was statistically unsignificant (*p* > 0.05). For all responses, the value of R^2^ was higher than 0.91, and the adjusted coefficient of determination (R^2^_adj_) was approximately 0.82. This indicates that the model exhibits good accuracy, showing that the percentage errors between the experimental and predicted values are less than 0.05. The interactions among four independent variables are depicted in the three-dimensional (3D) surface diagram shown in Figure 2g,h. The effective terms in the model equation obtained for TFC were the linear terms of A, B, C, and D, and the quadratic terms of A^2^, B^2^, C^2^, and D^2^ (Table 1). To have a good idea of the effects of four independent variables on the TFC yield, a quadratic model was established as follows:YTFC = 65.38 − 3.97A − 2.06B + 3.18C + 1.79D − 4.03A^2^ − 9.56B^2^ − 2.74C^2^ − 3.67D^2^
(2)
where YTFC represents the yield of TFC. A, B, C, and D represent the four independent variables of ultrasonic power, ethanol concentration, SLR, and extraction time, respectively. According to the F and *p* values in Table 1, the influencing order of extraction conditions on TFC are ultrasonic power >SLR > ethanol concentration > extraction time, and this shows that all four parameters have significant effects on the yield of TFC. Moreover, the significant differences in TFC yields were determined by the linear and quadratic factors.

### 2.3. Model Validation

The effectiveness of the proposed model was evaluated by comparing the predicted values obtained from running the prediction equation of the response surface model. The optimal parameters were determined by maximizing the response values of both models (TPC and TFC) simultaneously. The optimal parameters were ultrasound power of 205.9 W, ethanol content of 46.4%, SLR of 1:31.7, and extraction time of 20.1 min; the highest values of TPC and TFC were 115.27 ± 0.76 mg GAE/g and 67.04 ± 1.34 mg RE/g, respectively. The relative errors of TPC and TFC were 3.20% and 4.84%, respectively, between the actual and predicted values (Table 1). Thus, these optimum parameters were reliable and accurate for establishing the extraction process of TPC and TFC. Under these optimized conditions, TPC values at PGS I, PGS II, and PGS III were 117.22 ± 0.55, 112.73 ± 1.68, and 107.33 ± 1.39 mg GAE/g, respectively, whereas the extract of PGS I had the highest TFC (68.48 ± 2.01 mg RE/g), followed by PGS II (63.20 ± 1.01 mg RE/g) and PGS III (59.66 ± 1.87 mg RE/g) (Table 2). These results show that the TPC and TFC yields from *L. similis* flowers are related to developmental stages. Reports on rose, peach, and patchouli showed similar changes, and the phenolic compounds content in tender flowers was apparently greater than that in other flowers [21,53,54]. These differences may be due to the content and composition of phenolic compounds in *L. similis* flowers in three harvest periods.

### 2.4. Composition Analysis of Phenolic Compounds

The antioxidant activity of extracts are usually attributed to their content and the composition of phenolic compounds, which are significantly related to harvest stages [55,56,57]. Table 3 shows the quantified results of phenolic compounds in the gained optimum extracts, and eight phenolic compounds with different contents were detected at three harvest stages. The order of the phenolic compounds content was as follows: chlorogenic acid > quercetin > rutin > ferulic acid > isoquercetin > epicatechin > caffeic acid > kaempferol. The content of chlorogenic acid was between 96.6 and 103.4 mg/g at three harvest stages, while the content of kaempferol ranged from 1.42 to 5.91 μg/g. These results show that the differential changes in eight phenolic compounds content are recorded at three harvest stages. As exhibited in Figure 3a, the contents of chlorogenic acid, quercetin, rutin, isoquercetin, and epicatechin at PGS I of LSF are higher than those of at PGS II and PGS III. It was noted that the contents of rutin, chlorogenic acid, epicatechin, and quercetin in the flowers of apple, rose, and *Helle borusniger* gradually decreased with flower development up to senescence stage [53,58,59]. The decrease in these phenolic compounds can be attributed to continuous consumption as part of the plant’s antioxidant defense mechanism during flowering development. However, the present study indicates that the contents of ferulic acid, kaempferol, and caffeic acid are contrary to those in previous reports (Figure 3a). For example, the ferulic acid content in apple flowers gradually increases during the flower’s development. The elevated levels of kaempferol in flowers may be due to its direct or indirect influence on reproductive processes [58,60]. These findings provide evidence that the differential content of phenolic compounds occurs in a developmental-specific manner and differential regulation occurs in response to the harvest stages, and displays certain differences in different plant species. These phenolic compounds have strong antioxidant activity, and their contents play a major role in monitoring the changes in the free radical scavenging rate [57,61]. Previous studies have demonstrated that the DPPH radical scavenging rate of ferulic acid is approximately 45% lower than that of caffeic acid under the same conditions [62]. Studies have also showed that the antioxidant capacity of ferulic acid and kaempferol is concentration-dependent. In the present study, the antioxidant activity of the extracts was highly correlated with the content of chlorogenic acid, rutin, and quercetin. Moreover, the antioxidant activity was negatively related to the levels of ferulic acid and kaempferol (Figure 3b). Thus, the above-mentioned results provide a comprehensive evaluation of the clear correlation between the phenolic compounds and antioxidant properties in LSF at three development stages, and the relationship of phenolic compounds and antioxidant properties is more complex than expected.

### 2.5. Antioxidant Capacity of Extracts

DPPH and ABTS radical scavenging ability have been extensively applied to assess the antioxidant ability of biological compounds in various plant species [63]. The present results suggest that the IC_50_ values of ABTS and DPPH scavenging activities of extracts at PGS I (0.248 mg/mL and 0.139 mg/mL, respectively) are higher than those at PGS II (0.269 mg/mL and 0.152 mg/mL, respectively) and PGS III (0.272 mg/mL and 0.158 mg/mL, respectively) (Figure 4a,b). This may be related to the differences in phenolic compounds contents and their composition at three harvest stages, which are shown in Table 3 and Figure 3. Moreover, the comparative analysis results of the antioxidant activity of extracts and dried samples show that the ABTS radical scavenging rates of solid extracts (over 80%) are significantly higher than those of liquid extracts (36.9–45.2%). The DPPH radical scavenging rate of liquid extracts can reach 43.3%, and the values of solid extracts were over 90% (Figure 4c). These results exhibit that the extracts and dried samples possess a strong antioxidant capacity, which is associated with antioxidative phenolic compounds.

### 2.6. FTIR Analysis

FTIR has been widely used to identify organic groups of plant phenolics in plant extracts, and provide some key information about the chemical valence structure characteristics of bioactive compounds. FTIR can also easily identify the stretching and bending vibrational bands of hydroxyl and carboxyl groups in an extract [64,65]. As shown in Figure 5, phenolic compounds from LSF at three harvest stages exhibit 11 similar characteristic absorption peaks, indicating the similar composition of phenolic components. Previous studies have indicated that the characteristic absorption peak of phenolic compounds in the collected optimum extracts were observed in the range of 3400–3200 cm^−1^, which were corrected to the combined stretching vibrations of hydroxyl groups [66,67]. The peaks near 2936 cm^−1^ represented the stretching vibrations of the aromatic C-H in the polyphenols extract [68], and near 1715 cm^−1^ and 1610 cm^−1^ for the vibration of the C=O structure and the extension of the benzene ring, respectively [69]. The peaks near 1350 cm^−1^, 1208 cm^−1^, and 1040 cm^−1^ correspond to the folding of -CH groups in the extract, the stretching of amino groups of C-N, and the vibration of C-C, respectively [69,70]. Moreover, the bands ranging from 769 to 659 cm^−1^ represented the out-of-plane bending vibration of O-H [71]. The present results indicate that the variation characteristics of FT-IR spectroscopy are consistent with the differences in phenolic compounds from LSF at three harvest stages.

### 2.7. SEM Analysis

As shown in Figure 6, SEM analysis shows the differences in particle size, morphology, and surface roughness of powder and extracts from LSF at three harvest stages. It is obvious that the flower powders at three harvest stages have many surface attachments and a high degree of agglomeration, which may help to accelerate the released components from the crushed flower tissue during the extraction process (Figure 6c,d). During ultrasound-assisted extraction, ultrasound can also promote the shedding of lumps from the particle surface, and further improve the extractability of TPC and TFC from plant samples [72,73]. As shown in Figure 6f,g, it is worth mentioning that the dried extracts from LSF at three harvest stages have a small surface area and attachments, but there is no agglomeration. Moreover, the solid surface of dried extracts from LSF at PGS I seems to have less protrusions and roughness, and the shape is more irregular than that at PGS II and PGSIII. These differences might come from the combination of phenolic compounds, carotenoids, some ions (Ca^2+^), starch, and other substances due to the form of hydrogen, complexed, and covalent bonds created during the drying process [74,75,76,77]. From these findings, it can be deduced that the morphological differences of flowers at different harvest stage may be related to the extractability of TPC and TFC in LSF. However, the relationships of harvest stages and the chemical composition of phenolic compounds in LSF extracts was relatively complex, which are necessary to provide a detailed study on the developmental variability of bioactive components in LSF.

## 3. Materials and Methods

### 3.1. Flower Collection and Chemicals

Based on the Biologische Bundesantalt, Bundes-sortenamt, and Chemische Industrie (BBCH) scales [78,79], LSF from three phenological growth stages, including young alabastrum (PGS I, 51–55), slightly white alabastrum (PGS II, 56–60), and silvery flower (PGS III, 61–65), were selected and collected in Xing-ma Town, Nanjiang County, Sichuan Province, China. These samples were dried at 55 °C and thoroughly crushed. The moisture content of sample materials at PGS I (78.58% ± 1.03%), PGS II (79.74% ± 1.34%), and PGS III (84.13% ± 0.93%) was recorded. The sample powders were filtered using an 80-mesh sieve. Samples were stored under strictly dry conditions at room temperature. The standards of rutin, kaempferol, epicatechin, caffeic acid (CFA), chlorogenic acid (CA), ferulic acid (FA), isoquercitrin, and quercetin with ≥98% purity were obtained from Shanghai Yuanye Biotechnology Co., Ltd. (Shanghai, China). Gallic acid, DPPH (2,2-diphenyl-1-picrylhydrazyl), vitamin C (L-ascorbic acid), Folin–Ciocalteu phenol reagent, ABTS (2,2′-azino-bis(3ethylbenzothiazoline-6-sulfonic acid)), and Trolox (6-hydroxy-2,5,7,8-tetramethylchroman-2-carboxylic acid) were obtained from Macklin Company (Shanghai, China). Methanol and acetonitrile were acquired from J & K Chemical Ltd. (Shanghai, China). Ultrapure water was prepared using a Milli-Q Purification system (Millipore, MA, USA). Other chemical reagents utilized were of analytical grade.

### 3.2. Single-Factor Experiments

For better describe the results, the total phenol content (TPC) and total flavonoids content (TFC) extracted from LSF were combined and referred to as LSF phenolic compounds (LSF-PC). LSF extract was investigated under various ultrasonic powers (50–500 W), ethanol concentrations (5–80%), solid–liquid ratios (SLRs) (1:15, 1:30, 1:45, 1:60, 1:75, and 1:90 g/mL), and extraction times (5–30 min). Each experiment was performed three times. As illustrated in Table 4, the single-factor experiment of LSF was designed and conducted under the following conditions to assess the impact of different variables on the extraction yield of LSF-PC.

### 3.3. Optimizing LSF-PC Extraction Based on the Box–Behnken Design (BBD)

Following the principles of the Box–Behnken design (BBD), the experiments were designed using the results obtained from the single-factor experiment. The LSF-PC was considered as the response variable, while the ultrasonic power (A), ethanol concentration (B), SLR (C), and extraction time (D) were considered as independent variables. Each parameter was varied at three levels (−1, 0, and 1). The levels of petameters and their interactions are depicted in Appendix A, containing 3 replicates of the center points in 27 groups.

### 3.4. Determination of LSF-PC

The TPC of LSF was estimated according to the modified method of Sharma et al. [80]. Briefly, samples of 400 μL were blended with 400 μL of FC chromogenic agent. Subsequently, 1.2 mL of 7.5% Na_2_CO_3_ and 2.0 mL of distilled water were added in sequence. These mixtures were placed stably for 60 min at room temperature in the absence of light. The OD values were recorded at 765 nm, and the results are shown as milligrams of gallic acid equivalent (GAE) per gram of dry weight (mg GAE/g DW). TPC standard curve: y = (x − 0.0017)/0.0055.

The TFC was quantified using Wang et al.’s method [81]. The extracts of 400 μL were absorbed and thoroughly mixed with 1.6 mL of 60% ethanol, and then 120 μL of 5% NaNO_2_ was added and left to stand for 6 min. Immediately after this, 120 μL of 10% AlCl_3_ and 1.6 mL NaOH (1 mol/L) were added in turn. Finally, 160 μL of distilled water was added to the above-mentioned mixtures, and then stewed 20 min. The OD510 values were estimated, and the results were calculated and shown as milligrams of rutin equivalent (RE) per gram of dry weight (mg RE/g DW) based on the rutin standard curve: y = (x − 0.0017)/0.0055.

### 3.5. Preparation of LSF Extracts

The extraction process of phenolic compounds from LSF was performed under optimum experimental parameters, followed by centrifugation at 10,000 rpm for 5 min at 4 °C, and the supernatants and residues were obtained. These extracts were filtered and concentrated using a rotary evaporator (RE-52A, Shanghai Yarong Instrument Co., Ltd., Shanghai, China) at 55 °C. The extracts and residues were dried in a vacuum-drying oven (DZF, Shanghai Longyue Instrument Equipment Co., LTD, Shanghai, China) at 50 °C.

### 3.6. HPLC Analysis

Nine phenolic compounds, namely luteolin, epigallocatechin, rutin, chlorogenic acid, quercitrin, ferulic, isoquercitrin, kaempferol, and caffeic acid, were identified and quantified using HPLC-MS. The contents of rutin, quercetin, luteolin, kaempferol, and isoquercitrin were separated and analyzed using Agilent 1100 HPLC (Agilent, Santa Clara, CA, USA) at a 360 nm wavelength. The contents of CA and FA were determined at a 325 nm wavelength, and epicatechin and CFA were determined at a 280 nm wavelength. The optimal extracts were treated by centrifugation at 12,000 rpm for 10 min at 4 °C, and the supernatant was harvested. A C18 reverse-phase chromatographic column (250 mm × 4.6 mm, 5 μm) was applied at a flow rate of 1 mL/min, an injection volume of 5 μL, and a column temperature of 30 °C. For rutin, quercetin, luteolin, kaempferol, epicatechin, CFA, FA, and CA, the mobile phase contained a 0.1% phosphoric acid aqueous solution and methanol. For isoquercitrin, the mobile phase contained a 0.1% phosphoric acid aqueous solution and acetonitrile. The quantification conditions and standard curves of nine phenolic compounds are shown in Appendix A.

### 3.7. Determination of Antioxidant Activity

#### 3.7.1. DPPH Radical Scavenging Capacity

The analysis of the DPPH radical scavenging capacity was assessed using Barros et al.’s method [82]. Briefly, the 2 mL DPPH reagent was mixed with 100 μL of the extract, followed by stewing for 30 min at room temperature in the absence of light. The OD517 values were recorded, and the Vc and green tea polyphenols (GTPs) were used as controls. The activities were calculated using the following formula:DPPH scavenging rate (%) = [1 − (A_a_ − A_b_)/A_a_] × 100% (3)
where A_b_ represents the OD517 value of the DPPH mixed with the extracts, and A_a_ represents the OD517 value of the DPPH mixed with deionized water. The results were recorded as the scavenging rate (%).

#### 3.7.2. ABTS Radical Scavenging Activity

The ABTS radical scavenging activity was assessed using Chen et al.’s method [83]. In brief, the ABTS solution was blended with 14 mM of ABTS solution and 4.9 mM of potassium persulfate solution at room temperature in the absence of light for 16 h. The created mixtures were diluted using phosphate buffer (pH = 7.34) up to the OD734 value of 0.70 ± 0.01, and the diluted solution was applied as an ABTS working solution. Subsequently, 2 mL of the prepared working solution was combined with 200 µL of the extracts, and a standstill time of no less than 6 min was employed at room temperature in the absence of light. The OD734 value (designed as A1) was measured, and the blank control contained 200 μL of deionized water (designated as A0). GTP and VC were applied as the positive controls. ABTS radical scavenging activity was calculated using the following formula:ABTS radical scavenging activity (%) = [1 − (A0 − A1)/A0] × 100% (4)

### 3.8. Analysis of Fourier Transform Infrared Spectroscopy (FT-IR)

The structure of extracts was examined using FT-IR (Thermo Nicolet iS5, Florence, SC, USA). The extracts were encapsulated in KBr particles and subjected to scanning ranging from 4000 to 400 cm^−1^.

### 3.9. Analysis of Scanning Electron Microscope (SEM)

The samples of flower powder, extracts, and residue were fixed onto the aluminum head using viscous double-sided ribbons, and sputtered to cover the gold layer in a vacuum. These samples were then observed using SEM (FEI-Quattro S, Beijing, China) operating at 5 keV.

### 3.10. Statistical Analysis

In this study, every experiment was conducted in triplicate, and all the data are presented as the mean ± standard deviation. Statistical analysis was performed using analysis of variance, and the observed differences were found to be statistically significant (*p* < 0.05). Statistical analysis was conducted using SPSS 26.0, while data visualization was performed using Origin 2022, Design Expert 13, power point 2019, and adobe illustrator 2023.

## 4. Conclusions

In summary, the present study comprehensively reported the extraction optimization, composition identification, and antioxidant properties of phenolic compounds from LSF at three harvest periods using the UAE procedure. The results show that the maximum TPC and TFC values at PGS I are 117.2 ± 0.55 mg GAE/g and 68.5 ± 2.01 mg GAE/g, respectively, under a sonication intensity of 205.9 W, sonication time of 20.1 min, ethanol concentration of 46.4%, and SLR of 1:31.7. The estimated models supplied efficient mathematical descriptions of the extraction yields of TPC and TFC from LSF at three harvest stages. Significant differences in TPC and the antioxidant activity of extracts at three harvest periods were found. Eight phenolic compounds with variable contents were identified in LSF, and chlorogenic acid, quercetin, and rutin were confirmed as the dominant phenolic compounds. Overall, this study provided data support for the optimum extraction of total phenolic and flavone compounds, as well as those from LSF at three harvest stages, which provided theoretical guidance for expanding the high-value utilization of phenolic compounds in food and pharmaceutical industries. However, this study still lacks a multi-aspect evaluation of phenolic compounds from LSF, and further knowledge about the purification of these bioactive compounds and other biological characteristics will be needed.

## Figures and Tables

**Figure 1 molecules-29-03280-f001:**
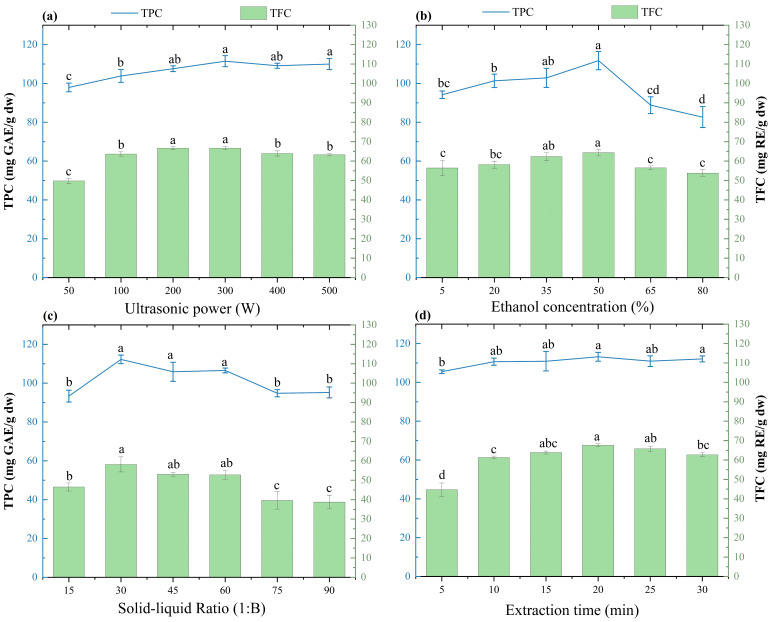
Influences of extraction factors on the yields of TPC and TFC from *L. similis* flowers. (**a**) Ultrasonic power, (**b**) ethanol concentration, (**c**) SLR, and (**d**) extraction time. Lowercase letters indicate significant differences among tested levels (*p* < 0.05).

**Figure 2 molecules-29-03280-f002:**
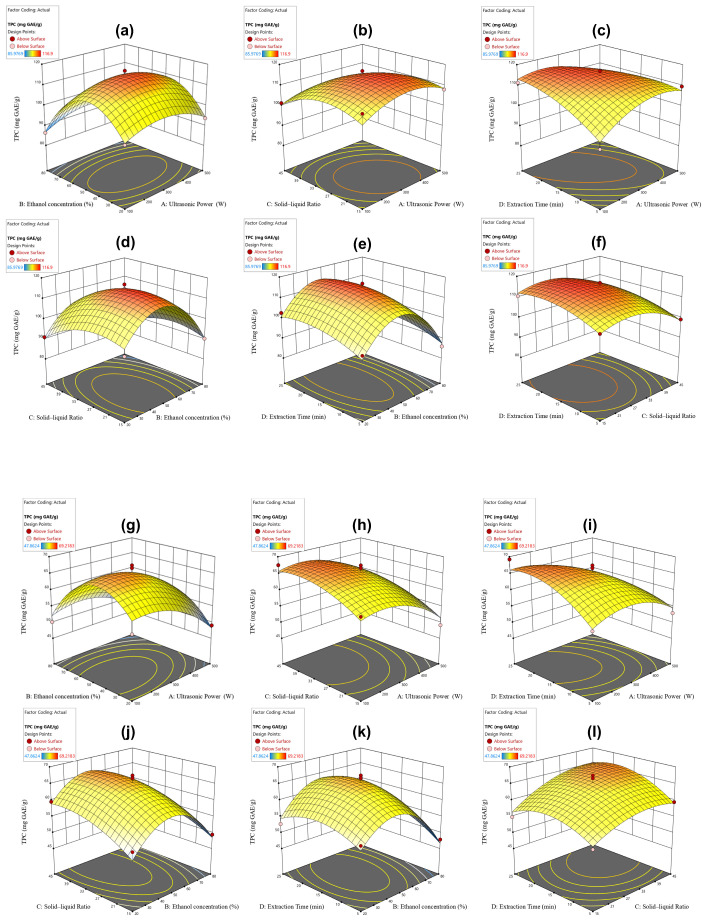
The influences of independent variables on the TPC and TFC yields from the *L. similis* flower by three-dimensional surface analysis. (**a**,**g**) Ultrasonic power versus ethanol concentration. (**b**,**h**) SLR and extraction time. (**c**,**i**) Ultrasonic power versus extraction time. (**d**,**j**) Ethanol concentration versus SLR. (**e**,**k**) Ethanol concentration and extraction time. (**f**,**l**) SLR and extraction time. Data represent mean ± SEM, *n* = 3.

**Figure 3 molecules-29-03280-f003:**
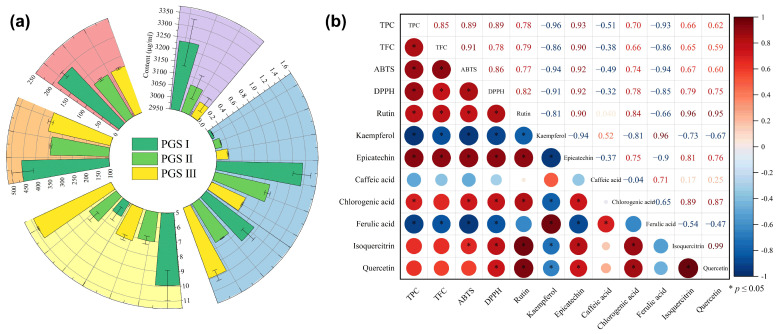
The quantitative results of eight phenolic compounds from the *L. similis* flower at three harvest stages (**a**), and the correlation analysis between phenolic compounds and antioxidant activity (**b**). Data represent mean ± SEM, n = 3.

**Figure 4 molecules-29-03280-f004:**
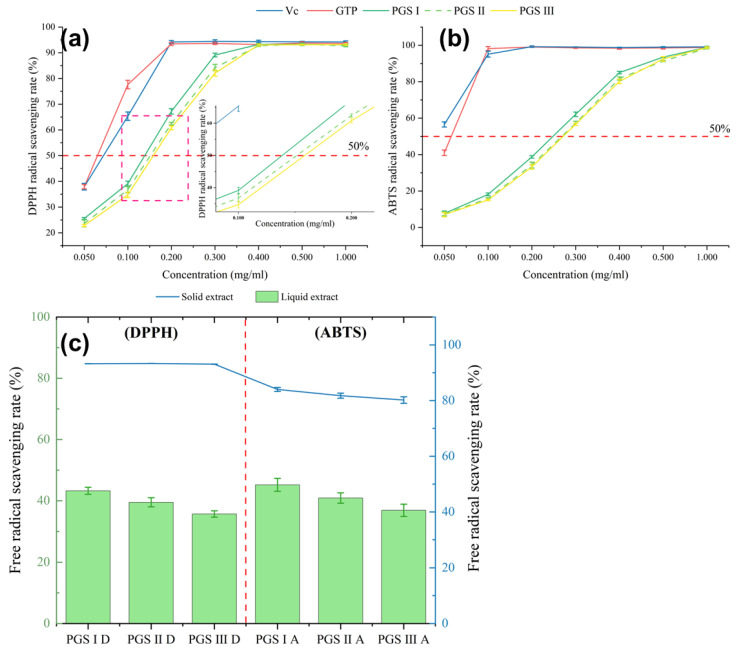
Antioxidant activity of phenolic compounds from the *L. similis* flower at three harvest stages. (**a**) DPPH radical scavenging capacity. (**b**) ABTS radical scavenging capacity. (**c**) Comparison analysis of antioxidant activities of DPPH and ABTS of solid extract and liquid extract. The concentration of liquid extract is 0.4 mg sample/mL. Data represent mean ± SEM, n = 3.

**Figure 5 molecules-29-03280-f005:**
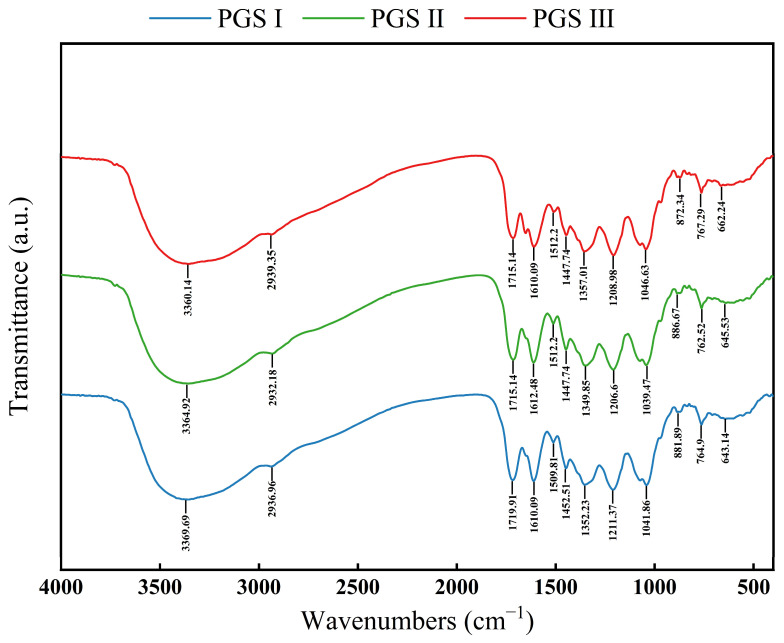
FTIR spectra of phenolic compounds from the *L. similis* flower at three harvest stages.

**Figure 6 molecules-29-03280-f006:**
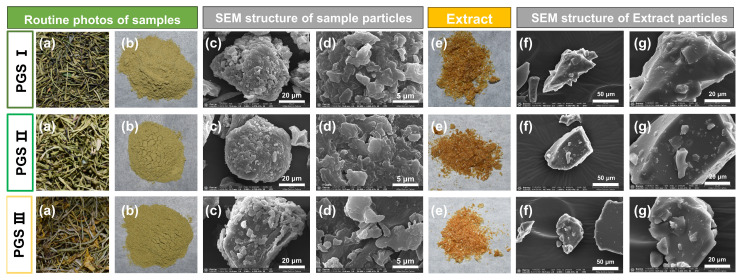
SEM analysis of sample powder and extracts from the *L. similis* flower at three harvest stages. Each row corresponds to a PGS, and each column corresponds to a letter. Conventional photographs of dried samples and powders (**a**,**b**). SEM pattern of sample particles; magnification: 5000× and 20,000 × (**c**,**d**). Solid crystal photograph of extract (**e**). SEM pattern of extract samples; magnification: 2000× and 5000× (**f**,**g**).

**Table 1 molecules-29-03280-t001:** ANOVA statistics of the quadratic model for the extraction yields of phenolic compounds from *L. similis* flowers. * indicates a significant mark, and the *p*-value less than 0.05; ** indicates that the *p*-value is less than 0.01; *** indicating that the *p*-value is less than 0.0001.

Source	df	TPC			TFC		
		SS	F-Value	*p*-Value	SS	F-Value	*p*-Value
**Model**	14	2223.96	15.39	<0.0001 ***	950.827	9.068	0.0002 **
**A**	1	0.3729	0.0361	0.8524	188.76	25.2	0.0003 ***
**B**	1	198.52	19.23	0.0009 **	50.9	6.8	0.0229 *
**C**	1	201.59	19.53	0.0008 **	121.11	16.17	0.0017 **
**D**	1	78.3	7.58	0.0175 *	38.49	5.14	0.0427 *
**AB**	1	5.33	0.5158	0.4864	10.15	1.36	0.2669
**AC**	1	13.21	1.28	0.2801	0.0104	0.0014	0.9709
**AD**	1	102.32	9.91	0.0084 **	27.6	3.69	0.6420
**BC**	1	3.2	0.3098	0.588	1.7	0.2274	0.0790
**BD**	1	0.1198	0.0116	0.916	10.81	1.44	0.2527
**CD**	1	0.1956	0.0189	0.8928	5.8	0.7739	0.3963
**A2**	1	191.79	18.58	0.001 **	86.61	11.57	0.0053 **
**B2**	1	1586.41	153.65	<0.0001 ***	487.36	65.07	<0.0001 ***
**C2**	1	204.67	19.82	0.0008 **	39.97	5.34	0.0395 *
**D2**	1	71.45	6.92	0.0219 *	71.65	9.57	0.0093 **
**Residual**	12	123.89			89.872		
**Lack of Fit**	10	116.46	3.13	0.2660	72.761	0.8504	0.6522
**Pure Error**	2	7.43			17.111		
**C. Total**	26	2347.85			1040.70		
**Std.Dev.**		3.21			2.7367		
**Mean**		100.19			56.491		
**C.V.%**		3.21			4.84		
**Adeq Precision**		12.8242			9.5148		
**R^2^**		0.9472			0.9136		
**Adjusted R²**		0.8857			0.8129		
**AICc**		191.40			182.73		

**Table 2 molecules-29-03280-t002:** TPC and TFC contents of the *L. similis* flower at three development stages under optimal conditions. Data represent mean ± SEM, n = 3. Different lowercase letters indicate significant difference at the 5% level.

	Ultrasound Power	Ethanol Content	SLR	Extraction Time	TPC Yield (mg GAE/g)	TFC Yield (mg RE/g)
**PGS** **I**	205.9 W	46.4%	1:31.7	20.1 min	117.22 ± 0.55 a	68.48 ± 2.01 a
**PGS** **II**	205.9 W	46.4%	1:31.7	20.1 min	112.73 ± 1.68 b	63.20 ± 1.01 b
**PGS** **III**	205.9 W	46.4%	1:31.7	20.1 min	107.33 ± 1.39 c	59.66 ± 1.87 c

**Table 3 molecules-29-03280-t003:** The identified and quantified results of phenolic compounds extracts from the *L. similis* flower at three harvest stages. Different lowercase letters indicate significant difference at the 5% level.

Phenols Compounds	Formula	Retention Time (min)			PGS I Content	PGS II Content	PGS III Content
		PGS I	PGS II	PGS III			
Rutin (mg/g)	C_27_H_30_O_16_	15.988	15.995	15.978	6.13 ± 0.31 a	3.91 ± 0.12 b	3.95 ± 0.09 b
Kaempferol (μg/g)	C_15_H_10_O_6_	20.454	20.473	20.471	1.42 ± 0.15 c	3.45 ± 0.19 b	5.91 ± 0.385 a
Epicatechin (μg/g)	C_15_H_14_O_6_	19.460	19.487	19.478	43.50 ± 4.50 a	28.30 ± 1.42 b	18.90 ± 0.37 c
Caffeic acid (μg/g)	C_9_H_8_O_4_	6.837	6.820	6.842	30.30 ± 3.73 b	22.90 ± 0.25 c	37.90 ± 2.80 a
Chlorogenic acid (mg/g)	C_16_H_18_O_9_	14.590	14.589	14.622	103.40 ± 3.13 a	98.10 ± 1.66 b	96.60 ± 1.05 c
Ferulic acid (mg/g)	C_10_H_10_O_4_	20.786	20.791	20.797	0.195 ± 0.02 c	0.23 ± 0.01 b	0.35 ± 0.001 a
Isoquercitrin (mg/g)	C_21_H_20_O_12_	8.297	8.270	8.251	0.32 ± 0.03 a	0.22 ± 0.01 b	0.23 ± 0.01 b
Quercetin (mg/g)	C_15_H_10_O_7_	12.912	12.860	12.832	14.60 ± 1.42 a	10.90 ± 0.14 c	11.60 ± 0.34 b

Data represent mean ± SEM, n = 3.

**Table 4 molecules-29-03280-t004:** Single-factor experiments of LSF-PC.

Experiment Factor	Ultrasonic Power (W)	Ethanol Concentration (%)	SLR	Extraction Time (min)
SLR	100	20	15–90	5
Ethanol concentration	100	5, 20, 35, 50, 65, 80	30	5
Ultrasonic power	100, 200, 300, 400, 500	20	30	5
Extraction time	100	20	30	5, 10, 15, 20, 25, 30

## Data Availability

The data presented in this study are available upon request from the corresponding authors.

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
