# Peer review of "Ultrasonic-Assisted Extraction of Phenolic Compounds from Lonicera similis Flowers at Three Harvest Periods: Comparison of Composition, Characterization, and Antioxidant Activity"

_molecules, 2024, doi:10.3390/molecules29143280_

Round 1
Reviewer 1 Report
Comments and Suggestions for Authors
The manuscript focuses on the optimisation of ultrasound-assisted extraction of phenolic compounds from Lonicera similis flowers at three harvest periods. Total phenolic and flavonoid contents, antioxidant activity and content of some phenolic compounds were determined. Moreover, the functional groups and microstructure of the gained extracts were assessed using Fourier transform infrared spectroscopy and scanning electron microscopy. The article brings new perspectives to the field and makes a contribution to the existing literature. The language and structure of the article are clear and easily understandable. The abstract provides a comprehensive overview of the study's objectives, methodology, and findings. Introduction provide sufficient background and include all relevant references. The results are clear.
The comments for the authors:
1. This sentence "These findings may provide useful references on the optimal extraction way of phenolic compounds in LSF at three harvest periods, which will help to capture higher phytochemical yield at optimal harvest stage." is strange and should be rewritten.
2. Why did the authors quantify only selected phenolic compounds?
3. Table 3. Last three columns should labeled as "content" with a unit. Unit should be the same for all the compounds. Statistical analysis should be provided.
4. Moisture content of sample material should be provided.
5. Calibration curve and correlation factor for TPC and TFC are missing.
6. Why did the authors choose DPPH and ABTS methods for determination of antioxidant activity?
Author Response
Comment 1. This sentence "These findings may provide useful references on the optimal extraction way of phenolic compounds in LSF at three harvest periods, which will help to capture higher phytochemical yield at optimal harvest stage." is strange and should be rewritten.
Response 1: We agree, thanks for these comments, that it has been changed to “These findings may provide useful references on the optimal extraction method for phenolic compounds in LSF at three different harvest periods, which will help to achieve a higher phytochemical yield at the optimal harvest stage (PGS â… ).”
Comment 2. Why did the authors quantify only selected phenolic compounds?
Response 2: Previous qualitative studies have identified the phenolic compounds of honeysuckle [1], and the selection of phenolic compounds in this study was based on their findings.
[1] Guo X M, Ma M H, Ma X L, et al. Quality assessment for the flower of Lonicera japonica thunb. during flowering period by integrating GC-MS, UHPLC-HRMS, and chemometrics[J]. Industrial Crops and Products, 2023, 191: 115938.
Comment 3. Table 3. Last three columns should labeled as "content" with a unit. Unit should be the same for all the compounds. Statistical analysis should be provided.
Response 3: OK. We agree. We have made some changes.
Comment 4. Moisture content of sample material should be provided.
Response 4: OK. We agree. The moisture content of sample material is PGS â… (78.58%±1.03%), PGS â…¡ (79.74%±1.34%), PGS â…¢ (84.13%±0.93%), we have added some descriptions in the revised manuscript as suggested.
Comment 5. Calibration curve and correlation factor for TPC and TFC are missing.
Response 5: OK. We agree. We have added some descriptions in the revised manuscript as suggested.
TPC: y=(x+0.0065)/0.0058
TFC: y=(x-0.0017)/0.0055
Comment 6. Why did the authors choose DPPH and ABTS methods for determination of antioxidant activity?
Response 6: Many studies have used DPPH and ABTS as evaluation criteria for antioxidant activity [2,3]. Compared with other test methods that assess antioxidant activity, DPPH and ABTS are considered to be more stable and accurate.
[2] Liu Z, Zhao M, Wang X, et al. Response surface methodology-optimized extraction of flavonoids with antioxidant and antimicrobial activities from the exocarp of three genera of coconut and characterization by HPLC-IT-TOF-MS/MS[J]. Food Chemistry, 2022, 391: 132966.
[3] Wu H, Zhao W, Zhou J, et al. Extraction, analysis of antioxidant activities and structural characteristics of flavonoids in fruits of Diospyros lotus L[J]. LWT, 2024: 116248.

Reviewer 2 Report
Comments and Suggestions for Authors
The paper shows that ultrasonic-assisted extraction is a powerful technique for extracting phenolic compounds from plants, offering advantages in efficiency, speed, and environmental sustainability.
The optimization of the extraction parameters has been carried out correctly. The paper is well-designed and supported.
Some minor questions and opinions.
Line 130 – separation between …were 111.48….
Line 148 – separation between …and 64.33…
Line 160 delete one to (toto)
The FTIR analysis, while used in many articles to add analysis data, often lacks the depth and specificity needed to enhance the value of the papers truly. So maybe not necessary.
In the SEM analysis, it is unclear why there are differences in the plants if the method of decreasing particle size is the same and the plant is the same, only harvested at different times.?? Can you give some more explanation?
The explanation may need more support, and the correlation with the content of TPC and TFC cannot be supported.
Author Response
Comment 1.
Line 130 – separation between …were 111.48….
Line 148 – separation between …and 64.33…
Line 160 delete one to (toto)
Response 1: OK. We agree, thanks for these comments. We have made changes.
Comment 2.
The FTIR analysis, while used in many articles to add analysis data, often lacks the depth and specificity needed to enhance the value of the papers truly. So maybe not necessary.
Response 2: We agree with your point of view, but for the integrity of the article, we suggest not deleting this part. We will pay attention to this issue in subsequent research and make corresponding changes.
Comment 3.
In the SEM analysis, it is unclear why there are differences in the plants if the method of decreasing particle size is the same and the plant is the same, only harvested at different times.?? Can you give some more explanation?
Response 3: The texture of LSF flowers varies at different stages: PGS â… is firmer, while PGS â…¢ is softer, potentially influencing the powder characteristics. Furthermore, during the PGS â… stage, bracts are present on the exterior of the flower buds. Post-flowering (PGS â…¡-PGS â…¢), these bracts degenerate, leaving only petals as the external structure. This transition further accounts for the differences.
Comment 4.
The explanation may need more support, and the correlation with the content of TPC and TFC cannot be supported.
Response 4: Thank you for mentioning this. We have made corresponding changes to the original text. This unsupported inference has been deleted.
